# Differentiation of Salivary Gland and Salivary Gland Tumor Tissue via Raman Imaging Combined with Multivariate Data Analysis

**DOI:** 10.3390/diagnostics14010092

**Published:** 2023-12-30

**Authors:** Miriam C. Bassler, Mona Knoblich, Elena Gerhard-Hartmann, Ashutosh Mukherjee, Almoatazbellah Youssef, Rudolf Hagen, Lukas Haug, Miguel Goncalves, Agmal Scherzad, Manuel Stöth, Edwin Ostertag, Maria Steinke, Marc Brecht, Stephan Hackenberg, Till Jasper Meyer

**Affiliations:** 1Process Analysis and Technology (PA&T), School of Life Science, Reutlingen University, Alteburgstr. 150, 72762 Reutlingen, Germany; miriam.bassler@reutlingen-university.de (M.C.B.); mona.knoblich@reutlingen-university.de (M.K.); ashutosh.mukherjee@reutlingen-university.de (A.M.); edwin.ostertag@rpt.bwl.de (E.O.); 2Institute of Physical and Theoretical Chemistry, Faculty of Science, University of Tübingen, Auf der Morgenstelle 18, 72076 Tübingen, Germany; 3Institute of Pathology, University of Würzburg, Josef-Schneider-Str. 2, 97080 Würzburg, Germany; elena.hartmann@uni-wuerzburg.de (E.G.-H.); almoatazbellah.youssef@uni-wuerzburg.de (A.Y.); lukas.haug@uni-wuerzburg.de (L.H.); 4Department of Oto-Rhino-Laryngology, Plastic, Aesthetic & Reconstructive Head and Neck Surgery, University Hospital Würzburg, Josef-Schneider-Str. 11, 97080 Würzburg, Germany; hagen_r@ukw.de (R.H.); goncalves_m@ukw.de (M.G.); scherzad_a@ukw.de (A.S.); stoeth_m@ukw.de (M.S.); hackenberg_s@ukw.de (S.H.); 5Chair of Tissue Engineering and Regenerative Medicine, University Hospital Würzburg, Röntgenring 11, 97070 Würzburg, Germany; maria.steinke@isc.fraunhofer.de; 6Fraunhofer Institute for Silicate Research ISC, Röntgenring 11, 97070 Würzburg, Germany

**Keywords:** salivary gland tumor, confocal Raman imaging, principal component analysis, discriminant analysis, multivariate data analysis, molecular diagnostics

## Abstract

Salivary gland tumors (SGTs) are a relevant, highly diverse subgroup of head and neck tumors whose entity determination can be difficult. Confocal Raman imaging in combination with multivariate data analysis may possibly support their correct classification. For the analysis of the translational potential of Raman imaging in SGT determination, a multi-stage evaluation process is necessary. By measuring a sample set of Warthin tumor, pleomorphic adenoma and non-tumor salivary gland tissue, Raman data were obtained and a thorough Raman band analysis was performed. This evaluation revealed highly overlapping Raman patterns with only minor spectral differences. Consequently, a principal component analysis (PCA) was calculated and further combined with a discriminant analysis (DA) to enable the best possible distinction. The PCA-DA model was characterized by accuracy, sensitivity, selectivity and precision values above 90% and validated by predicting model-unknown Raman spectra, of which 93% were classified correctly. Thus, we state our PCA-DA to be suitable for parotid tumor and non-salivary salivary gland tissue discrimination and prediction. For evaluation of the translational potential, further validation steps are necessary.

## 1. Introduction

Salivary gland tumors (SGTs) are a sub-group of head and neck tumors and account for 3% to 6% of all head and neck neoplasms [1,2]. Most SGTs are of benign nature, representing approx. 80% of all incidences [3,4]. More than thirty different malignant and benign SGTs are known according to the 2017 WHO classification, of which the two main benign tumors are the cystadenolymphoma (Warthin tumor) and pleomorphic adenoma [5]. Due to the high variety in tumor entities, a reliable diagnosis only based on histo- or cytomorphological criteria is difficult and sometimes impossible [6]. As a result, pre- and intraoperative malignancy assessment based on fine-needle aspiration cytology, core needle biopsy and open biopsy is sometimes unreliable [7,8,9]. However, reliable malignancy assignment is necessary to avoid revision surgery [10]. The cytological methods frequently used preoperatively and also intraoperative histopathological diagnostics highly depend on the experience of the pathologist and may occasionally be subject to uncertainties, even in the case of experienced diagnosticians [6].

In addition to standard cyto- and histopathologic approaches, spectroscopic methods in combination with multivariate data analysis (MVA) are finding increasing acceptance to support tumor diagnostic methodologies [11,12,13,14,15,16,17]. By applying different spectroscopy principles, either chemical or morphological signatures are obtained, which reveal tissue-related features and are thus highly tissue-specific. Due to improvements in spatial resolution and scanning speed, spectroscopic imaging became increasingly popular for data acquisition [18,19,20]. Consequently, large imaging data sets with a high spatial and spectral resolution are obtained [21,22]. The linkage with MVA techniques, such as principal component analysis (PCA), allows the extraction of the most relevant spectral characteristics from the imaging data and to distinguish the samples based on the previously extracted attributes [23,24]. A combination of PCA with discriminant analysis (DA) additionally describes the distinction in a quantitative manner and enables a prediction of non-included spectra or samples. Typical imaging methods applied for tumor diagnosis are Fourier-transform infrared spectroscopy [25], fluorescence [26] and Raman imaging [27], which are mainly associated with statistical tools, such as PCA, DA, support vector machine, the k-nearest neighbor algorithm or artificial neural network analysis [25,28,29,30]. The spectroscopy-based models are used for the identification or distinction of brain [31], colon [32], breast [33] or head and neck tumors [34,35].

First spectroscopic attempts of parotid tumor identification were performed using Raman spectroscopy coupled with support vector machine [36,37]. Further studies focused on identifying differences in lipid composition and changes in secondary protein structure between cancerous and non-cancerous salivary gland tissue using Raman imaging [38]. Our group just recently published a study employing Raman spectroscopy and PCA analysis to enable a differentiation between benign pleomorphic adenoma and low-malignant adenoid cystic carcinomas [39]. Although primary efforts in differentiating SGTs were achieved, no study on deploying spectral differences has been published yet to implement a Raman-MVA model as diagnostic adjunct and review its functionality.

To evaluate the translational potential of Raman imaging (RI) for the overarching objective to use RI for supporting SGT entity and malignancy determination, probably a multi-stage data analysis process will be necessary. Due to the high number of tumor entities of SGTs, a multi-stage process with identification of the spectra that were acquired in the tumor tissue and exclusion of all spectra that were acquired in the non-tumor salivary gland tissue could help to enhance the statistical accuracy. Therefore, our study deals with the implementation of a PCA-DA model using Raman imaging data of normal salivary gland tissue, Warthin tumor and pleomorphic adenoma to differentiate between the tissue types and predict unknown parotid samples. Relevant tissue regions of unstained parotid cross-sections are chosen based on a hematoxylin and eosin (HE) assessment and measured using Raman imaging. The obtained imaging data are initially analyzed via a detailed Raman band evaluation. Afterwards, a PCA is computed to achieve the best possible tissue type separation due to tissue-specific information and combined with a DA to enable the classification of model-unknown Raman spectra. By applying the final PCA-DA, new tissue areas of interest are tested and classified in a proof-of-principle concept. The assignments are validated by a corresponding HE diagnosis to confirm whether the PCA-DA predictions are correct. Final model classification abilities are expressed by the overall accuracy, sensitivity, specificity and precision. As a result of a successful PCA-DA implementation, its utilization as diagnostic assistance in the clinical routine might be possible.

## 2. Materials and Methods

### 2.1. Patient Selection and Parotid Tissue Sample Preparation

This study was approved by the institutional ethics committee on human research of the Julius-Maximilians-University Würzburg (vote 224/18). All experiments were performed according to the Declaration of Helsinki. All patients agreed to participate in this study through informed consent.

Patients with a salivary gland tumor (parotid tumor) were preoperatively screened, and respective tissue specimens were selected. Suitable tissue samples had to consist of either Warthin tumor (*n* = 5) or pleomorphic adenoma (*n* = 4) and non-tumor salivary gland tissue (*n* = 9) in order to be included in the final sample set. All tissues were identified by a trained pathologist.

Tissue specimens were initially cut into smaller tissue pieces, which were arranged to display tumor and non-tumor salivary gland tissue. Afterwards, the tissue pieces were frozen in this arrangement and a series of 10 µm thick consecutive cryosections was prepared. Parotid cryosections were placed onto quartz objective slides (Suprasil^®^ 1, Aachener Quarzglas-Technologie Heinrich GmbH & Co. KG, Aachen, Germany) and subsequently fixed with a 4% paraformaldehyde solution (ROTI^®^Histofix 4%, Carl Roth, Karlsruhe, Germany) for 30 min. Following cross-section fixation, three washing steps with phosphate buffered saline were performed, and the samples were dried overnight at ambient temperature. These unstained sections were used for Raman imaging.

A detailed histopathological evaluation was applied based on corresponding 3 µm thick HE-stained cross-sections. The HE staining was performed according to a standardized protocol. HE cross-sections were analyzed and imaged using the BZ-9000 BIOREVO System (Keyence, Neu-Isenburg, Germany). Whole slide images were scanned at 40× using a slide scanner, Panoramic SCAN II (3DHISTECH, Budapest, Hungary).

### 2.2. Confocal Raman Imaging

Raman data acquisition was performed with a confocal microscopic setup (Alpha 300 RA&S, WITec, Ulm, Germany) according to the summarized measurement and data extraction workflow illustrated in Figure 1.

A confocal microscope system, equipped with a lens-based spectrometer (UHTS 300, WITec, Ulm, Germany) and a CCD camera (DU970, Andor Technology, Belfast, UK), was used for Raman imaging [40]. Excitation was performed with a 532 nm frequency-doubled Nd:YAG laser transmitted via a single-mode fiber onto the tissue cross-sections. The inelastically scattered light was collected with a 20× objective (EC Epiplan, 20×/0.4, Carl Zeiss AG, Oberkochen, Germany) and transferred to the spectrometer and CCD camera (EMCCD, 16 Bit, 1600 × 200 pixel, 16 µm × 16 µm, thermoelectrically cooled: −60 °C) via a 100 µm core diameter multimode fiber. Each Raman image encompassed a size of 20 µm × 20 µm and was acquired with a scan step size of 1 µm and a scan speed of 30 s/line. Thus, 20 × 20 spectra within the image were recorded with an integration time of 1.5 s per spectrum, resulting in a total number of 400 Raman spectra per tissue area. The Raman shift was recorded from 50 cm^−1^ to 3670 cm^−1^ with a 600 g/mm (blaze wavelength (BLZ) = 500 nm) grating and centering the spectrometer at 2000 cm^−1^. The spectral resolution of the optical system was 2 cm^−1^ [41]. Overall, three tumorous areas (Warthin tumor or pleomorphic adenoma) and three salivary gland areas per tissue sample and patient were selected for Raman imaging. Following image acquisition, Raman spectra were subtracted from the quartz background and treated with a baseline fluorescence correction (shape: 100) and a cosmic ray correction (filter size: 2, dynamic factor: 8).

### 2.3. Data Pre-Treatment and Multivariate Data Analysis

MVA was performed with The Unscrambler X 10.5 (Camo Analytics AS, Oslo, Norway). All model-included Raman mean spectra were preprocessed equally by applying a standard normal variate transformation followed by a Savitzky–Golay smoothed 1st order derivative (2nd polynomial order, symmetrical 41 points). The spectral area of 900–1750 cm^−1^ was used for MVA calculations, while additional spectrum ranges (50 cm^−1^ to 900 cm^−1^, 1750 cm^−1^ to 3670 cm^−1^) without relevant information were excluded for variable reduction. The PCA was calculated with mean centering, a full cross-validation and the singular value decomposition algorithm to distinguish between the different parotid tissue types. Model outliers were identified in the influence plot hotelling’s T^2^ versus F-residuals (outlier limits 5% each) and excluded from the model if proven to be true.

PCA was combined with DA by using the Mahalanobis distance algorithm and PCA score values. The number of deployed principal components (PCs) for the DA was similar to the shown PCA model. In total, 5 PCs were applied for calculating the DA. The overall accuracy, sensitivity, specificity and precision of the PCA-DA model were additionally calculated based on the confusion matrix terminology [42,43].

## 3. Results

### 3.1. Raman Mean Spectra Analysis

To reveal significant spectral differences, Raman mean spectra for normal salivary gland, Warthin tumor and pleomorphic adenoma tissues were compared and calculated by averaging all respective Raman spectra to one overall mean spectrum including the corresponding 95% confidence interval, illustrated in Figure 2. Potential differences between the spectra can be a varying number of Raman bands, a changing relation between bands or different intensities or variations in shape. From 500 cm^−1^ to 3500 cm^−1^ (Figure 2a), no clear spectral variations are distinguishable. Two major bands at 746 cm^−1^ and 2927 cm^−1^ correspond to a ring breathing of DNA/RNA bases and to CH/CH_3_ vibrations of lipids and proteins, respectively, and are identical in each spectrum. A detailed band analysis in the range of 900–1700 cm^−1^ was performed, and main spectral differences were highlighted (Figure 2b, (1)–(14)). A superimposed illustration of this range is shown in Appendix A, Appendix A and a summary of all Raman signal assignments is presented in Table 1.

At 915 cm^−1^, one distinct weak Raman band (1) occurs for normal salivary gland tissue, which is less pronounced or even absent for both parotid tumors (Figure 2b), and can be assigned to vibrations of ribose RNA (Table 1). Instead, pleomorphic adenoma tissue reveals a band at 920 cm^−1^ that can be ascribed to a C-C stretching of collagen proline, which is also visible in the Warthin tumor, but far less distinctive. A further spectral band at 1043 cm^−1^ (2) is clearly illustrated for the salivary gland tissue, but missing for both tumors that can also be assigned to collagen proline (Figure 2b, Table 1). Varying intensity relations can be observed for two bands at 1158 cm^−1^ and 1168 cm^−1^ (3). In case of salivary gland tissue, the band at 1158 cm^−1^ is more highlighted than the second band at 1168 cm^−1^, which is opposite for the Warthin tumor and pleomorphic adenoma tissues. Both Raman bands can mainly be deduced from C-N and C-C stretching vibrations of proteins and C-C/C=C lipid oscillations (Table 1). While salivary gland and pleomorphic adenoma tissues exhibit a Raman band at 1197 cm^−1^ (4), the Warthin tumor exposes a slightly shifted band maximum at 1204 cm^−1^ (4, Figure 2b). The bands can either be allocated to antisymmetric phosphate (1197 cm^−1^) or amide III and CH_2_ wagging vibrations (1204 cm^−1^) of glycine backbone or proline side chains (Table 1). Changes in band intensity, shape and position between the tissue types are indicated for the 1222 cm^−1^ band (5). It is most intense for salivary gland tissue, but varies in intensity and shape for Warthin tumor tissue and is even slightly shifted for pleomorphic adenoma tissue (1228 cm^−1^) (Figure 2b). The two signals (1222 cm^−1^ and 1228 cm^−1^) are mainly assignable to amide III vibrations of proteins and thymine/adenine stretching of DNA/RNA (Table 1). Between 1300 and 1400 cm^−1^, a series of distinct Raman bands (6–9) occurs in all three Raman mean spectra with pronounced shoulders or double maxima mainly for Warthin tumor tissue (6–8), but also for salivary gland (9) and pleomorphic adenoma (8) tissues, respectively. Reasons for the occurrence of these Raman bands are oscillations of lipids, collagens, proteins, tryptophan and β-carotene within the respective tissues (Table 1). Additional differences in intensity relations emerge between two Raman band maxima at 1443 cm^−1^ and 1454 cm^−1^ (10), which reveal a slightly more emphasized first maximum at 1443 cm^−1^ in the salivary gland and Warthin tumor spectra compared to the second band at 1454 cm^−1^. In contrast, pleomorphic adenoma tissue shows equally dominant Raman bands for both (Figure 2b). The two bands are mainly caused by the CH_2_/CH_3_ stretching of proteins, lipids, triglycerides and collagen (Table 1). A Raman band at 1517 cm^−1^ (11) is observed for salivary gland tissue with the highest intensity, followed by Warthin tumor and pleomorphic adenoma tissues with a decreasing peak appearance. This band is ascribable to the C-C stretching mode of β-carotene (Table 1). The Raman band at 1585 cm^−1^ (12) solely exhibits an additional shoulder in the Warthin tumor spectrum, which is missing in the salivary gland and pleomorphic adenoma spectra and corresponds to a C=C olefinic stretching of proteins (Table 1). Signature differences between the parotid tissues can further be attributed to Raman bands at 1628 cm^−1^/1640 cm^−1^ (13) and 1654 cm^−1^ (14), which mainly result from the C_α_=C_α_, C-C and C=O stretching modes of proteins, collagen and lipids (Table 1). Aside from these few spectral differences (1)–(14), the Raman patterns are highly similar and overlapping, with small signatures that can hardly be assigned to distinct vibrational modes. Therefore, a PCA evaluation is inevitable to extract minor pattern variations as a result of the high spectral similarity.

### 3.2. Raman Data Analysis via PCA-DA

Due to there being only minor Raman band differences in the overall mean spectra, an improved discrimination of all three tissue types is needed. This was accomplished by a PCA calculation. As a PCA allows the extraction of the most relevant and differing chemical information of the Raman spectra, a precise distinction of the tissue types is enabled. For this purpose, three tumorous areas (Warthin tumor or pleomorphic adenoma) and three non-tumor salivary gland tissue areas were selected for Raman image acquisition, which resulted in 6 × 400 Raman spectra (2400 spectra) for each patient. Consequently, 9 × 6 × 400 Raman spectra (21,600 spectra) were measured overall. Mean spectra were formed by averaging 100 adjacent, processed spectra, which yielded four spectra per tissue area. Therefore, a total number of 216 Raman mean spectra (48 pleomorphic adenoma, 60 Warthin tumor and 108 salivary gland tissue) were used to generate the PCA model. The final PCA is illustrated in Figure 3. Different perspectives of the 3D PCA scores plot are shown in Appendix A, Appendix A. Since the most dominant changes were previously identified within 900–1750 cm^−1^, PCA was performed only on the basis of this region.

The PCA model encompasses a total number of five PCs to distinguish between the different parotid gland tumors and the parotid gland tissues and reaches a total explained variance of 77%, considering PC1–PC5. The number of PCs was chosen based on the explained variance and the interpretable information on loadings [58,59,60]. The aim was to create a PCA-DA model with as few PCs as possible for a reliable prediction of the parotid tissue types. Information about the relation of the calibration and validation PCA is given in Appendix A. Within the 3D scores plot (Figure 3a), PC1, PC2 and PC5 are presented, explaining 40%, 18% and 4% of the explained variance, respectively. The separation of the Warthin tumor cluster (blue triangles) has already been realized by PC1 and PC2. Here, the Warthin cluster is arranged almost completely below-average for PC1 and above-average for PC2. Additionally, PC1 and PC2 cause a group splitting of the salivary gland (green circles) and pleomorphic adenoma (red squares) cluster into mixed tissue type groups, respectively. Here, salivary gland and pleomorphic adenoma are randomly organized into above and below the average of PC1 and PC2, with no apparent cluster formation. This is more clearly shown in the 2D scores plot illustration in Appendix A. Therefore, PC5 is required to clearly demarcate between salivary gland and pleomorphic adenoma spectra. Within the PC2/PC5 plane, the salivary gland cluster is completely structured below-average, whereas the pleomorphic adenoma group is organized above-average in terms of PC5 (Figure 3a). The Warthin group, however, is arranged above-average for PC2 and almost completely below-average for PC5 within this setting. As a result, a clear group separation between Warthin tumor, pleomorphic adenoma and salivary gland spectra is achieved, with only minor cluster overlaps at the center (Figure 3a). Although PC3 and PC4 explain 15% of the model variance, this represented information does not contribute to the tissue type differentiation. An overview of the complete PCA model including all PCs (PC1–PC5), as well as related 2D scores and loading plots, is represented in Appendix A.

In Figure 3b, the corresponding loading plots of PC1, PC2 and PC5 are illustrated. The loadings plots display the highest spectral influence on the tissue type separation in the 3D scores plot. Some of the main influencing maxima can even be referred to Raman bands assigned in Figure 2. Negative maxima at 1164 cm^−1^ (3, Figure 2, Table 1) and 1530 cm^−1^ in the PC1 loadings have a major impact on the tissue separation (Figure 3b). Further influencing maxima can be defined at 1596 cm^−1^ and 1643 cm^−1^ (13, Figure 2, Table 1) according to the PC1 loadings. The PC2 loadings plot shows the most dominating impact between 1440 and 1680 cm^−1^ with main positive maxima at 1440 cm^−1^ (10, Figure 2, Table 1) and 1643 cm^−1^ (13, Figure 2, Table 1) and negative maxima at 1467 cm^−1^ and 1680 cm^−1^ (Figure 3b). Within the PC5 loadings, a few pronounced positive maxima can be determined at 1450 cm^−1^ (10, Figure 2, Table 1) and 1546 cm^−1^ as well as negative ones at 1285 cm^−1^ and 1384 cm^−1^.

Based on the formed PCA model, a DA was subsequently calculated by using the PCA score values of PC1 to PC5 and the Mahalanobis distance algorithm. The PCA-DA’s performance is evaluated using an internal validation process. For this purpose, each model-included spectrum was allocated to one tissue cluster as if it was not contained in the model. As a result, a confusion matrix is obtained (Appendix A), which describes the accordance of the model’s assignments and the pathologist’s diagnosis and is used to compute the model’s accuracy, sensitivity, specificity and precision (Table 2). Salivary gland model spectra were most accurately assigned with 97% accuracy, whereas Warthin tumor spectra resulted in 91% correctly matched spectra. Pleomorphic adenoma is still allocated correctly with 89% accuracy. Due to the excellent group assignment of the model spectra, high performance parameters of 94% accuracy, 94% sensitivity, 95% specificity and 94% precision were accomplished (Table 2).

The PCA-DA was finally used to predict model-unknown Raman mean spectra of all investigated parotid tissues in order to verify the model’s classification abilities. For this purpose, histopathologically distinct regions of non-tumor salivary gland tissue (Figure 4, 1–5, light blue squares) and tumor tissue (Figure 4, 1–5, black squares) were previously defined as suitable prediction regions on HE sections (Figure 4, 1). These HE regions were afterwards identified on the consecutive, unstained cross-section used for Raman imaging (Figure 4, 2; 1–5, light blue and black squares). After Raman image acquisition, mean spectra of the tissues were again calculated by processing and averaging 100 adjacent Raman spectra (Figure 4, 3 and 4). This second group of 15 Raman mean spectra (5 per entity; 1500 spectra) was predicted by the final PCA-DA model and evaluated whether they were classified falsely (red cross) or correctly (green check mark) (Figure 4, 5). This corresponded to an external model validation. The model prediction outcome was finally corroborated by the initial HE diagnosis, which allowed a final assessment of the PCA-DA. Typical HE sections of a pleomorphic adenoma (Figure 4f) and a Warthin tumor (Figure 4g) with a magnification of 4× are shown in detail to compare the different morphological features.

The classification results of unknown Raman mean spectra by the PCA-DA model are summarized in Table 3. Almost all Raman mean spectra were assigned to the correct tissue type, except for one pleomorphic adenoma spectrum, which was classified as salivary gland tissue.

## 4. Discussion

Accurate classification of salivary gland tumors can be difficult but is clinically very important. Therefore, new approaches that can support a correct diagnosis are of great interest. Thus, we focused on Raman spectroscopic imaging combined with MVA to create a valid and robust PCA-DA model for the differentiation of salivary gland, Warthin tumor and pleomorphic adenoma tissues.

A comparison of Raman mean spectra in the range of 900–1700 cm^−1^ enabled us to reveal individual spectral patterns and thus spectral changes between the different tissue types (Figure 2b, Table 1). Throughout the investigated Raman shift, most of the identified Raman bands match between the tissue types with only a few distinct band differences (1)–(14), (Figure 2b). Most of these differences allow the discrimination of non-tumor salivary gland tissue from parotid gland tumors. This is expressed by several bands or band relations, which are different for normal salivary gland tissue than for both tumors. A distinctly appearing band (1) for the salivary gland tissue, which is less pronounced or shifted within the tumor spectra, indicates variations in RNA. Comparably, band (5) also implies changes in DNA/RNA between the tissue types. DNA/RNA differences can be correlated to the abundance of DNA/RNA in tumors due to an intensified proliferation and metabolic activity of cancer cells [61,62]. Aside from DNA/RNA related differences, mainly protein and lipid associated vibrations (3)–(5) allow a demarcation of non-tumor salivary gland tissue from Warthin tumor and pleomorphic adenoma (Figure 2b). This can be attributed to a dysregulated synthesis of certain lipids and proteins, often enhanced in tumors [63,64]. Additionally, characteristic tumor signal relations are observed at (3), (10), (13) for Warthin tumor and pleomorphic adenoma that are also related to triglyceride, lipid and protein variations and help to discriminate between tumor and non-tumor tissue. These band relations additionally enable distinguishing between the tumor entities themselves, as shown, e.g., for (13). This is assumed to result from the individual protein or lipid expression levels of both tumors [65]. Furthermore, the triglyceride indication can be ascribed to a high endogenous synthesis of fatty acids, which is estimated to be linked to the high degree of tumor cell proliferation [66]. Further bands, e.g., (1; 920 cm^−1^), (2), (10), (14), enable differentiating salivary gland tissue from parotid tumors (Figure 2b), which are expected to correlate with collagen vibrations of the tumor’s extracellular matrix. Collagen was proven to be increasingly accumulated and expressed in tumorous tissue and thus contribute to tumor progression, invasion and metastasis [67,68]. As a result, the abundance, type and composition of collagen might be different in the three tissue types, which also allows the distinction of the parotid tumors.

Although spectral differences are already indicated by the band allocations, the smallest changes that contribute to a clear parotid differentiation might not be assignable or might even be missed. To identify these, a PCA analysis between 900 and 1750 cm^−1^ was performed (Figure 3). The final tissue type separation was achieved by PC1, PC2 and PC5 and can be deduced from the corresponding loading plots (Figure 3b). The main loading maxima allow spectral regions or bands with the highest impact on the tissue demarcation to be identified. As the Warthin tumor is almost fully separated by PC1 and PC2, the main influences in both loadings are assumed to affect this segregation. Within the PC1 loadings, maxima at 1164 cm^−1^ (3, Figure 2, Table 1), 1530 cm^−1^, 1596 cm^−1^ and 1643 cm^−1^ (13, Figure 2, Table 1) are identified to have a major impact on the Warthin cluster position. The impact at 1164 cm^−1^ is associated with a tyrosine variation in the tissues that mainly causes a separation of Warthin tumor tissue from pleomorphic adenoma. Tyrosine was found to be accumulated as crystals in pleomorphic adenoma, assumed to result from a disordered protein synthesis [69]. This seems not to be the case for other parotid tumors, such as the Warthin tumor, which explains the discrimination of both tumor entities at 1164 cm^−1^ (3, Figure 2, Table 1) and also the differentiation from salivary gland tissue. Another PC1 loading maximum corresponds to 1530 cm^−1^, which can be correlated to a carotenoid vibration. Carotenoids were mainly shown to suppress tumor metastasis or progression, but were also suggested to promote tumor invasion by actively interfering into various signal pathways [70]. Variations in carotenoids in Warthin tumor tissue might thus result in a differently up- or downregulated gene expression, which affects changes in molecular mechanisms and tissue composition. As a consequence, the Warthin tumor group is more distinguishable from salivary gland and pleomorphic adenoma tissues on PC1. Furthermore, the influence of PC1 loadings at 1596 cm^−1^ and 1643 cm^−1^ (13, Figure 2, Table 1) are mostly ascribable to amide III oscillations of proteins and lipids that were previously discussed to be highly abundant in tumors as a result of protein and lipid dysregulation [63,64]. Consequently, this contributes to a segregation of the tumors from non-tumor salivary gland tissue. A high concentration of unsaturated triacylglycerol in healthy salivary gland tissue and a high protein concentration in SGT tissue were observed by Paluszkiewicz et al. [6].

PC2 is additionally needed to achieve a complete separation of the Warthin tumor group. Here, the loading at 1440 cm^−1^ (10, Figure 2, Table 1) enables a separation of Warthin tumor and pleomorphic adenoma from salivary gland tissue and is tentatively assigned to a cholesterol vibration. This can be explained by an increased cholesterol concentration in tumors, especially in proliferating cancer cells [71], which was shown for numerous cancers [71,72]. Particularly in Warthin tumors, the cystic structures contain cell debris and cholesterol crystals, which further supports the cholesterol-based explanation. Another explanation for the 1440 cm^−1^ band derives from a varying lipid composition or concentration in Warthin tumor tissue, as it is a characteristic vibration for lipids [73,74]. This might be corroborated by the fact that further cholesterol bands, such as 701 cm^−1^ and 1087 cm^−1^, are less pronounced in the Raman spectra. Additional PC2 loadings at 1467 cm^−1^, 1643 cm^−1^ (13, Figure 2, Table 1) and 1680 cm^−1^ again indicate a high impact of proteins and lipids on the separation of the Warthin tumor cluster, which implies a varying protein and lipid composition for Warthin tumors.

PC5, however, was required to enable a complete distinction between pleomorphic adenoma and normal salivary gland tissue. The most influencing PC5 loading maxima are shown at 1285 cm^−1^, 1384 cm^−1^, 1450 cm^−1^ (10, Figure 2, Table 1) and 1546 cm^−1^. The PC5 loading at 1285 cm^−1^ points to the impact of collagen on the distinction between pleomorphic adenoma and salivary gland tissue. This again confirms that tumors differ from normal tissues by their collagen composition, type and abundance, as was already discussed. Another influence is also assigned to cytosine, which is associated with tumorigenesis due to cytosine methylation [75]. Compared to that, pronounced PC5 loadings at 1384 cm^−1^ and 1450 cm^−1^ (10, Figure 2, Table 1) result from CH_2_/CH_3_ bending modes that are at least partly malignant-specific. A further assignment of these loadings is possible due to the CH_2_/CH_3_ modes of biomolecules, including proteins and lipids. Both loading allocations can be interpreted as an increase in biomass, typically observed for tumors [76], and are thus highly specific for pleomorphic adenoma. At a PC5 loading value of 1546 cm^−1^, bound/free nicotinamide adenine dinucleotide hydride (NADH) and tryptophan are expected to additionally trigger the separation of pleomorphic adenoma from salivary gland tissue. This is reasonable since tumors reveal an increased metabolism that correlates with a higher NADH and tryptophan demand [62,77]. Furthermore, the ratio of bound to free NADH differs between normal and tumorous tissue and thus contributes to a better distinction between non-tumor salivary gland tissue and pleomorphic adenoma [78].

The final PCA-DA model was used to predict model-unknown Raman mean spectra of the three tissue types and test its classification suitability. Model performance parameters are all beyond 90% and thus demonstrate the model’s applicability for prediction purposes. Except for one misclassification of pleomorphic adenoma, all Raman mean spectra were predicted correctly according to the corresponding HE diagnosis (Table 3). A possible reason for the failed prediction might be the high tissue heterogeneity of pleomorphic adenoma, which could hamper its classification by the model. However, all other pleomorphic adenoma spectra were distinctly identified by the model and thus suggest that the PCA-DA covers the variations in tissue heterogeneity. This indicates the high potential of the PCA-DA model to be a supporting diagnostic tool. Our findings support, that Raman spectroscopy (RS)-based imaging systems are able to separate non-tumor salivary gland tissue and salivary gland tumor tissue based on spectroscopic information with a high accuracy. However, further validation steps are necessary to evaluate whether RS-based spectroscopy approaches are suitable for salivary gland tumor entity determination. In the next step, the investigation of a data set with more different tumor entities is planned. Although a good classification accuracy is achieved, the time-consuming Raman imaging prevents the measurement of a large number of patient samples. Consequently, an automatization is required to enable a high imaging throughput in the future.

## 5. Conclusions

This study focused on the identification and differentiation of non-tumor salivary gland tissue, Warthin tumor and pleomorphic adenoma via Raman imaging combined with MVA. Representative regions of all three tissue types were Raman-imaged in a point-scanning manner with an adequate spatial and high spectral resolution. This allowed us to conduct a thorough Raman band analysis in order to identify differences in the spectra and understand the biological background. Spectral variations were mainly observed for RNA/DNA-, lipid-, protein- and collagen-related bands or band relations that primarily contribute to a discrimination of tumorous and non-tumorous tissues. Due to the high spectral similarity between the tissues, a subsequent PCA calculation with the Raman data was necessary to uncover the tissue-specific impact responsible for a distinct differentiation. This revealed major influences of tyrosine, carotenoids, cholesterol, cytosine, NADH and tryptophan, aside from proteins and lipids, that allowed for a clear parotid tissue separation via PCA. These results emphasized the necessity of PCA to extract the hidden information, not determinable to this extent by a Raman band analysis. By combining the PCA with a subsequent DA, a reliable PCA-DA model was formed, which achieved an accuracy of 94%, sensitivity of 94%, specificity of 95% and precision of 94%. To verify the model’s prediction capability, an external proof-of-principle validation of model-unknown Raman mean spectra was performed. The results revealed an almost completely correct prediction outcome, except for one false pleomorphic adenoma classification. Possible reasons for that were ascribed to its pronounced heterogeneity. All prediction results were additionally confirmed by corresponding HE diagnoses. Consequently, we consider our PCA-DA model to be a useful supportive means in identifying salivary gland tumor tissue. However further evaluation steps are necessary for a more in-depth assessment of the translational potential. In detail, an evaluation of the PCA-DA model with a big dataset containing a higher number of different salivary gland tumor entities is necessary.

## Figures and Tables

**Figure 1 diagnostics-14-00092-f001:**
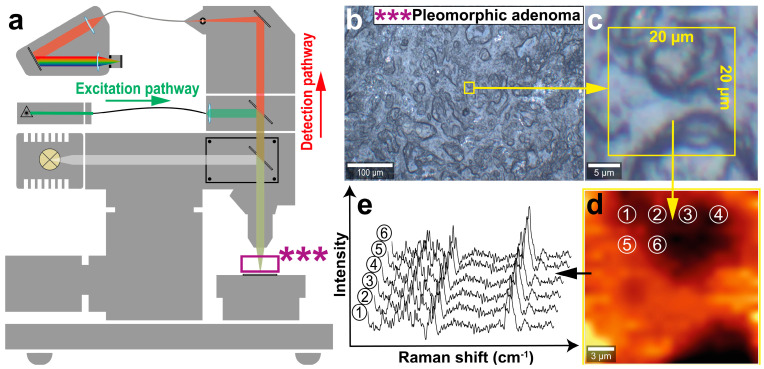
Schematic acquisition workflow for Raman imaging of parotid tissue cross-sections and data extraction. Raman images were recorded with the WITec Alpha 300 RA&S using a 532 nm frequency-doubled Nd:YAG laser for excitation purposes (**a**). Initially, a suitable tissue region was chosen in an overview brightfield image, which was in accordance with the corresponding HE evaluation ((**b**), *** marks the sample). In this overview image, a 20 µm × 20 µm tissue area of interest ((**b**,**c**), yellow box) was determined and Raman imaged in a stepwise manner. The obtained Raman image (**d**) is composed of single Raman spectra (1–6) located at distinct x,y-positions. Raman spectra were extracted individually for PCA-DA model formation (**e**).

**Figure 2 diagnostics-14-00092-f002:**
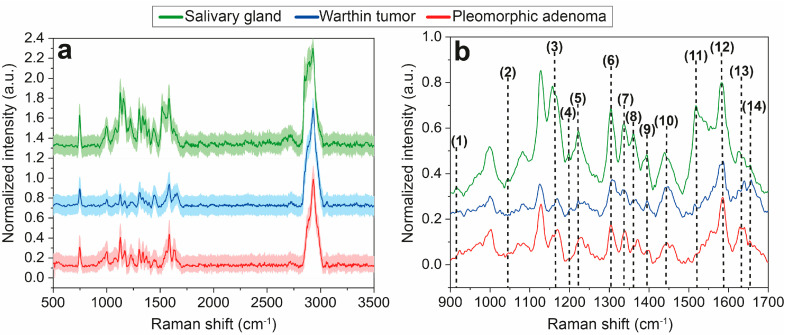
Overall Raman mean spectra with corresponding 95% confidence intervals (light colors) of normal salivary gland tissue (green), Warthin tumor (blue) and pleomorphic adenoma (red), illustrated in a Raman shift region of 500–3500 cm^−1^ (**a**). Main spectral differences were noticed between 900 and 1700 cm^−1^, which was analyzed in more detail (**b**). In this region, distinct Raman band assignments were performed (1–14). The spectra are vertically displayed. Corresponding molecular vibrations and causes are comprehensively listed in Table 1.

**Figure 3 diagnostics-14-00092-f003:**
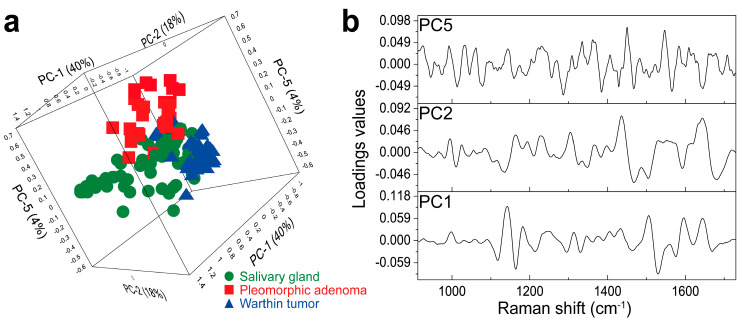
PCA model for the differentiation of salivary gland, Warthin tumor and pleomorphic adenoma. The 3D scores plot is illustrated in (**a**), displaying PC1 (40%), PC2 (18%) and PC5 (4%). A successful separation of salivary gland (green circles), Warthin tumor (blue triangles) and pleomorphic adenoma (red squares) clusters is achieved with only minor group overlaps in the center. The corresponding loading plots are shown in (**b**), revealing the main influencing Raman shift of PC1, PC2 and PC5 for the group segregation. To demonstrate the clear differentiation between parotid tissue clusters by the PCA model, various perspectives on the 3D scores plot are shown in Appendix A. Additional 2D scores and respective loading plots of all model-included PCs are summarized in Appendix A.

**Figure 4 diagnostics-14-00092-f004:**
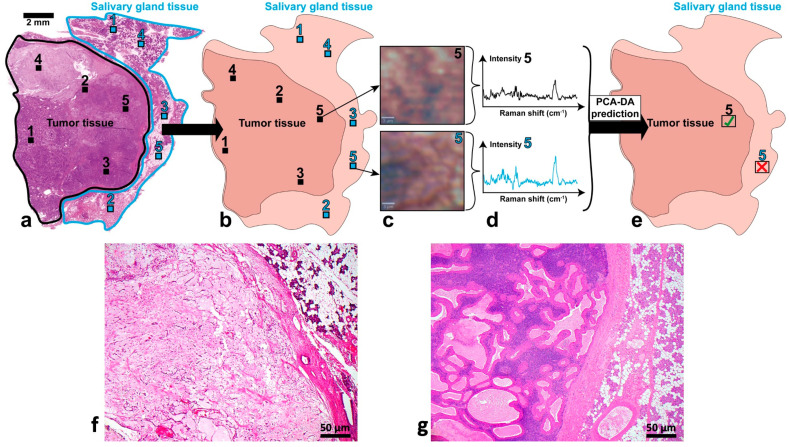
Schematic description of the external PCA-DA validation. At first, histopathologically appropriate tissue regions were defined on the HE-stained cross-sections (1–5 tumor tissue, 1–5 salivary gland tissue in (**a**)). Afterwards, these regions were identified on the corresponding unstained cross-sections used for Raman imaging (1–5 tumor tissue, 1–5 salivary gland tissue in (**b**)). Raman images (**c**) and resulting Raman mean spectra (**d**) were acquired and calculated to be predicted by the PCA-DA model. The tested Raman mean spectra are either correctly (✓) or falsely (✕) classified by the model (**e**), which is evaluated by the initial HE diagnosis in step (**a**). Examples for the typical histomorphology of (**f**) a pleomorphic adenoma and (**g**) a Warthin tumor with adjacent healthy salivary gland tissue.

**Table 1 diagnostics-14-00092-t001:** Raman band assignments for non-tumor salivary gland tissue, Warthin tumor, pleomorphic adenoma. Column 1 indicates the number assigned to the corresponding Raman band in the overall mean spectra (Figure 2b). Column 2 reveals the distinct relative shift (cm^−1^) of each Raman band identified in the mean spectra. The molecular vibration (column 3) resulting from specific molecules (column 4) within the tissues are listed. In column 5, all references of the Raman band assignments are summarized.

Number	Raman Shift/cm^−1^	Assignment	Cause	Reference [44]
(1)	915; 920	-; C-C stretch	Ribose RNA; collagen	[45,46]
(2)	1043	ring stretching	Collagen proline	[46]
(3)	1158; 1168	C-N stretching; ν(C=C); ν(C-C)	Proteins/lipids	[47,48]
(4)	1197; 1204	Antisymmetric phosphate vibrations; amide III, CH_2_ wagging vibrations	Glycine backbone or proline side chains	[48,49]
(5)	1222; 1228	C-N stretching and N-H bending, thymine, adenine stretch	Amide III, proteins, DNA/RNA	[47,50]
(6)	1304	CH_2_/CH_3_ deformation, twisting or bending	Lipids, collagen	[46,51]
(7)	1337	CH_2_/CH_3_ wagging, twisting and/or bending mode	Collagens, lipids, amide III (proteins)	[45,48,52]
(8)	1360	-	Tryptophan	[44]
(9)	1396	-	β-carotene	[44]
(10)	1443; 1454	CH_2_ deformation; CH_2_ stretching/CH_3_ asymmetric deformation	Lipids, proteins, triglycerides (fatty acids); elastin, collagen, phospholipids	[51,53,54,55]
(11)	1517	C-C stretch mode	β-Carotene accumulation	[53]
(12)	1585	C=C olefinic stretch	Proteins	[54,56]
(13)	1628; 1640	C_α_=C_α_ stretch; -	Proteins; amide I (proteins)	[44,57]
(14)	1654	C-C stretch, C=O stretching mode	Amide I (proteins), collagen, lipids	[46,48,51]

**Table 2 diagnostics-14-00092-t002:** Model performance parameters for the tissue entity distinction. All model-included Raman mean spectra (column 2) were ascribed to one tissue group in order to validate the PCA-DA. The total and relative (%) number of correctly assigned model spectra are listed in columns 3 and 4. Overall accuracy, sensitivity, specificity and precision of the model are summarized in columns 5–8.

Entity	Total Spectra	Correctly Assigned	Correctly Assigned/%	Accuracy /%	Sensitivity /%	Specificity /%	Precision /%
Salivary gland tissue	94	91	97	94	94	95	94
Pleomorphic adenoma	35	31	89				
Warthin tumor	47	43	91				

**Table 3 diagnostics-14-00092-t003:** Prediction outcome of model-unknown Raman mean spectra. Five Raman mean spectra for salivary gland, Warthin tumor and pleomorphic adenoma (column 1 and 2) tissues were classified, respectively, by the PCA-DA model (column 3) and compared with the initial HE diagnosis (column 4). Correctly predicted Raman mean spectra are indicated in green and bold, whereas wrong classifications are depicted in red and regular font.

Entity	Prediction Spectra	Model Classified as	HE Diagnosed as
Salivary gland tissue	1	**Salivary gland tissue**	**Salivary gland tissue**
	2	**Salivary gland tissue**	**Salivary gland tissue**
	3	**Salivary gland tissue**	**Salivary gland tissue**
	4	**Salivary gland tissue**	**Salivary gland tissue**
	5	**Salivary gland tissue**	**Salivary gland tissue**
Pleomorphic adenoma	1	**Pleomorphic adenoma**	**Pleomorphic adenoma**
	2	**Pleomorphic adenoma**	**Pleomorphic adenoma**
	3	Pleomorphic adenoma	Salivary gland tissue
	4	**Pleomorphic adenoma**	**Pleomorphic adenoma**
	5	**Pleomorphic adenoma**	**Pleomorphic adenoma**
Warthin tumor	1	**Warthin tumor**	**Warthin tumor**
	2	**Warthin tumor**	**Warthin tumor**
	3	**Warthin tumor**	**Warthin tumor**
	4	**Warthin tumor**	**Warthin tumor**
	5	**Warthin tumor**	**Warthin tumor**

## Data Availability

The datasets used and/or analyzed during the current study are available from the corresponding author on reasonable request.

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
