# Peer review of "Differentiation of Salivary Gland and Salivary Gland Tumor Tissue via Raman Imaging Combined with Multivariate Data Analysis"

_diagnostics, 2023, doi:10.3390/diagnostics14010092_

Round 1

Reviewer 1 Report

Comments and Suggestions for Authors

The manuscript entitled “Differentiation of Salivary Gland and Salivary Gland Tumor Tissue by Raman Imaging Combined with Multivariate Data Analysis” represents the interesting results in identifying salivary gland tumor tissue using Raman spectroscopy. However, in my opinion before publication some results should be included and a more detailed description of chemometrics should be provided.

1) The study should be significantly improved by citing the latest literature published within the last 5 years.

2) Why did you use a 20x objective? Was there any comparative analysis of Raman tissue spectra taken from 10x, 20x, and 50x for example?

3) I think it is need to add the confidence interval of spectral characteristics in Fig. 2.

4) It is noted in Lines 167 the cross-validation (CV) is used. However, paper do not contain detailed information about CV. I recommend adding results of CV in text.

5) It is not clear, why 5 PC are used to build PCA model. What rule or criterion was applied to select the optimal number of PCs? There are several rules to determine the correct number of PC to avoid the overfitting model (https://arxiv.org/abs/2210.10051, B. Boehmke, B.M. Greenwell, Hands-On Machine Learning with R). Loadings of PC4 and PC5 seems as noise (they look like a “fence”) and probably they do not contain useful information on the chemical composition of the tested samples. Although the PC4 and PC5 increase accuracy, they must be statistically and physically significant to exclude their random and unstable contribution to model. In my opinion, based on the presented data adding PC4 and PC5 in PCA model can lead to model overfitting. I also recommend adding RMSE of the classification model for training, cross validation cases, and test sets .

Author Response

Dear Reviewer 1,

thank you very much for your valuable suggestions. Please, see the attached file.

on behalf of the authors,

Miriam Bassler and Mona Knoblich

Reviewer 2 Report

Comments and Suggestions for Authors

The authors present the use of Raman imaging coupled with multivariate analysis, namely, PCA-DA to differentiate the salivary gland, Warthin tumour and pleomorphic adenoma in pathological tissue samples. The authors presented mean Raman spectra from the key regions of interest before acquiring Raman images from the selected areas and performing multivariate analysis to discriminate these regions. The methods appear to be reasonable; my only concern being that the area selected for Raman mapping represents a relatively small cross section of the whole tissue sampling, which could lead to the potential for biased sampling. If the authors were acquiring full Raman spectra across the range 50 cm-1 to 3670 cm-1, then the imaging rate could potentially be increased by acquiring only partial Raman spectra (as the region >1800 cm-1 was not used beyond Figure 1). That being said, the results appear to be sound and the classification model works appropriately well albeit on a relatively small sample scale. To that end, I agree with the authors’ conclusion that a larger data set containing a greater number of tissue samples is required. I have highlighted some major comments below for the consideration of the authors:

11)     In the discussion, the authors mention the detection of cholesterol in the Wartharin samples at 1440 cm-1. This is a typical lipid band for α(CH2/CH3) observed in many lipid species (10.1002/jrs.4607), and so I am not convinced by the interpretation here. Were other cholesterol modes detected (e.g. 701 cm-1 and 1087 cm-1) that have been observed before (10.1016/j.cmet.2014.01.019)?

22)     In Figure 3, The authors mention the separation of the Wartharin tumour cluster. However, this observation is not obvious in the current figure and presentation style. The separation across all PCA analyses in Fig S2 is challenging. Is it possible to re-orient the figure to show this separation more clearly?

33)     In Figure 5, the Wartharin tumour tissue sample is not shown. This should be included for reference, as the data points for the other two tissue types is available.

Minor comments:

Page 2 line 64: allows to extract the most… (grammar)

Page 2 line 81: no study concentrated yet on deploying spectral differences (grammar)

Page 7 line 247: As the PCA allows to extract (grammar)

Figure 3a: A separation of the Warthin tumor cluster (blue triangles) is already realized by 262 PC1 and PC2 (re-orientate the figure for clarity – the data points are overlapping).

Comments on the Quality of English Language

Generally very good, a few minor comments noted above.

Author Response

Dear Reviewer 2,

thank you very much for your valuable suggestions. Please, see the attached file.

on behalf of the authors,

Miriam Bassler and Mona Knoblich

Round 2

Reviewer 1 Report

Comments and Suggestions for Authors

I thank authors for their comments. I have no additional questions.

Reviewer 2 Report

Comments and Suggestions for Authors

The authors have responded to my concerns. One minor grammatical error in Line 286 as noted below.

Line 286: "create a PCA-DA model with less PCs as possible for a reliable prediction of the parotid". Replace "less PCs" with "as few PCs".

Comments on the Quality of English Language

No major concerns.